

# Anomalous mobility edges
# in one-dimensional quasiperiodic models

Tong Liu[1]⋆, Xu Xia[2]⋆, Stefano Longhi[3,4] and Laurent Sanchez-Palencia[5]

**1** School of Science, Nanjing University of Posts and Telecommunications,
Nanjing 210003, China
**2** Chern Institute of Mathematics and LPMC, Nankai University, Tianjin 300071, China
**3** Dipartimento di Fisica, Politecnico di Milano, Piazza L. da Vinci 32, I-20133 Milano, Italy
**4** IFISC (UIB-CSIC), Instituto de Fisica Interdisciplinar y Sistemas Complejos,
E-07122 Palma de Mallorca, Spain
**5** CPHT, CNRS, Ecole Polytechnique, IP Paris, F-91128 Palaiseau, France

⋆ These authors contributed equally to this work.

## Abstract

Mobility edges, separating localized from extended states, are known to arise in the single-particle energy spectrum of disordered systems in dimension strictly higher than two and certain quasiperiodic models in one dimension. Here we unveil a different class of mobility edges, dubbed anomalous mobility edges, that separate energy intervals where all states are localized from energy intervals where all states are critical in diagonal and off-diagonal quasiperiodic models. We first introduce an exactly solvable quasi-periodic diagonal model and analytically demonstrate the existence of anomalous mobility edges. Moreover, numerical multifractal analysis of the corresponding wave functions confirms the emergence of a finite energy interval where all states are critical. We then extend the sudy to a quasiperiodic off-diagonal Su-Schrieffer-Heeger model and show numerical evidence of anomalous mobility edges. We finally discuss possible experimental realizations of quasi-periodic models hosting anomalous mobility edges. These results shed new light on the localization and critical properties of low-dimensional systems with aperiodic order.

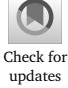

# 1 Introduction

The concept of *mobility edge* (ME), separating localized from nonlocalized phases, is central to the Anderson localization realm [1,2]. While arbitrarily weak disorder is sufficient to localize all the wavefunctions in dimension one (1D) or two, an energy threshold to localization (ME) appears in the spectrum of systems in dimension strictly higher than two [3]. The ME is characterized by multifractal wavefunctions, which are neither exponentially localized nor fully extended [4–6]. Anderson localization [7] has now been demonstrated in a variety of experiments, using photonic systems [8–10], acoustic waves [11], ultracold atoms [12–16], and quantum wires [17]. However, a direct experimental observation of MEs remains a challenge [18–22].

Quasiperiodic systems offer an appealing intermediate between periodically ordered and fully disordered systems, and various models have already been realized in experiments with ultracold atoms [23–27], photonic crystals [28], and polariton condensates [29,30]. While some engineered disorder correlations feature effective MEs in low dimensions [31–36], quasiperiodic systems allow for true localization transitions in 1D. The most considered case is the paradigmatic 1D Aubry-André (AA) model, which displays a localization transition at a critical amplitude of the quasiperiodic potential [37,38]. The AA model is, however, characterized by a special self-dual symmetry, which prevents the existence of a ME, and all the states in the spectrum suddenly change from extended to localized at the critical point [39–43]. Such features persist even when considering some non-Hermitian extensions [44–47]. Another widely studied case is the Maryland model, in which the quasiperiodic potential is unbounded [48–50]. In this case, however, (almost all) the eigenstates are exponentially localized and the Maryland model does not display a localization transition nor a ME. In other models, such as the Fibonacci chain, all the eigenstates are critical and there is no ME either [40,51].

Great interest has been devoted to find low-dimensional quasicrystals with MEs, including special incommensurate potentials with generalized AA self-duality [47,52,53], shallow multichromatic potentials [25,47,54–59], flat-band lattices [60], quasiperiodic mosaic lattices [61], and quasiperiodic pseudo-particle models [62]. Recently, it has been shown that long-range hopping can induce an energy threshold between extended and multifractal states [63]. In

this case, however, localization is destroyed.

Here we show that (short-range) quasiperiodic models can give rise to an energy interval of critical states while stabilizing localization in other energy intervals, hence inducing unconventional MEs, here dubbed *anomalous mobility edges* (AMEs). We first demonstrate mathematically the existence of such AMEs for an exactly-solvable diagonal model and corroborate this prediction using multifractal analysis within numerical calculations. We then show that AMEs can also appear in a nondiagonal but local quasiperiodic model. In the absence of an exact mathematical solution, we rely exclusively on numerical simulations and show evidence of the onset of AMEs separating critical energy intervals from localized energy intervals. Our results shed new light on localization and critical properties of aperiodic media, showing the existence of new classes of MEs in systems with properly designed quasiperiodic correlations. Possible implementations of the models considered here are discussed, including Floquet-engineered classical and quantum systems to emulate unbounded incommensurate potentials.

## 2 AME in an unbounded quasiperiodic potential: Mathematical proof

To prove the existence of AMEs, consider first a tight-binding model with nearest-neighbor hopping and quasiperiodic on-site potential, defined by the eigenvalue equation

$$E\psi_n = \psi_{n+1} + \psi_{n-1} + v(2\pi\alpha n + \theta)\psi_n \,, \tag{1}$$

where $\psi_n$ is the wavefunction amplitude at the lattice site $n$, $E$ its energy, and the hopping amplitude is set to unity. The function $v$ is periodic, $v(x+2\pi) = v(x)$, $\theta$ is a phase, and $\alpha$ is an irrational Diophantine number. A typical choice is the inverse golden number, $\alpha = (\sqrt{5}-1)/2$, which we adopt here. The AA model corresponds to the choice $v(x) = V\cos(x)$, with $V$ the quasiperiodic amplitude [37,38]. It displays a localization transition at $V = 2$ but no ME: For $|V| < 2$, all the wave functions are extended while for $|V| > 2$ they are all exponentially localized. This is a well known consequence of the self-dual symmetry [38]. In contrast, almost any self-duality-breaking potential $v$ induces a standard ME, separating a localized energy interval from an extended energy interval at some critical energy [25,47,52–58].

Consider now an unbounded potential $V_n = v(2\pi\alpha n + \theta)$, which, however, does not diverge at any lattice site $n \in \mathbb{Z}$. Any classical particle would be trivially localized in between two sites where the potential $V_n$ exceeding the particle energy. In contrast, for a quantum particle, tunneling allows for leaks. The Simon-Spencer theorem, however, states that, absolutely continuous spectra – and thus extended states – are forbidden [64]. It suggests that the wavefunctions may be localized. It is indeed so in the Maryland model, corresponding to the potential $v(x) = V\tan(x)$, which dislays a pure point spectrum with only exponentially localized states for any $V \neq 0$, see refs. [48,50]. Below, we however show that this is not always the case and that an appropriate choice of the potential $v(x)$ allows for a critical energy interval.

Consider for instance the potential

$$v(x) = \frac{V}{1 - a\cos(x)} \,, \tag{2}$$

where $V$ is the potential strength and $a$ a tuning parameter. This model has been studied in the bounded case, $0 < a < 1$, in ref. [53] and recently emulated with ultracold atoms [65]. In this case, the model displays a generalized AA self-dual symmetry and standard MEs appear. Here we focus on the unbounded case, $a > 1$, where the self-duality argument breaks down.

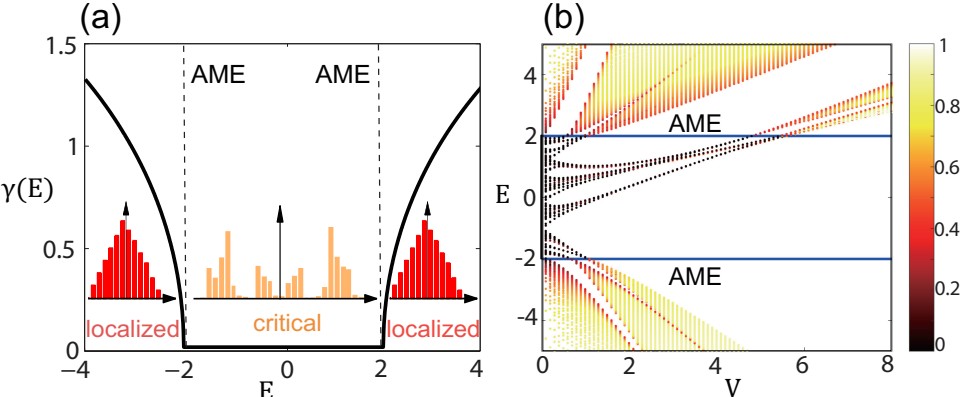

Figure 1: Anomalous mobility edge in an unbounded quasiperiodic potential. (a) Lyapunov exponent versus energy for the model of Eqs. (1) and (2), as found analytically [Eq. (5)]. The Insets show cartoons of localized and critical states in the corresponding phases. (b) Inverse participation ratio IPR versus potential amplitude and energy for the same model, as found using numerical calculations. The solid blue lines show the AMEs at $E_m = \pm 2$.

Setting $\theta \notin \pm[\arccos(1/a) + 2k\pi - 2\pi\alpha\mathbb{Z}]$ ensures that the potential is unbounded but finite at any lattice site $n$. We compute the Lyapunov exponent (LE, inverse localization length), $\gamma = 1/\xi$, using Avila's global theory for unbounded quasiperiodic operators [66–68]. It reads as

$$\gamma_\epsilon(E) = \lim_{n\to\infty} \frac{1}{2\pi n} \int_0^{2\pi} \ln\|T_n(\theta + i\epsilon)\| d\theta\,, \tag{3}$$

where $\|T_n(\theta + i\epsilon)\|$ is the norm of the transfer matrix, given by the ordered product

$$T_n(\theta) = \prod_{l=0}^{n-1} \begin{pmatrix} E - \nu(2\pi\alpha l + \theta) & -1 \\ 1 & 0 \end{pmatrix}. \tag{4}$$

Note that the complexification of the phase ($\theta \to \theta + i\epsilon$) plays a crucial role. The calculation may then be performed analytically by factorizing out the unbounded term and taking the limit $\epsilon \to 0$ (see Appendix A). It yields

$$\gamma(E) = \max_\pm \left( \ln\left| \frac{E \pm \sqrt{E^2 - 4}}{2} \right| \right). \tag{5}$$

A plot of the LE versus energy is shown in Fig. 1(a). Remarkably, $\gamma(E)$ is independent of the potential parameters $V$ and $a$, provided $a > 1$. For $|E| > 2$, we find $\gamma(E) > 0$ and for $\alpha$ a Diophantine number, we conjecture that, like for other quasiperiodic models, the LE provides the asymptotic ($n \to \pm\infty$) exponential decay rate of the wave function. Hence, for $|E| > 2$, the spectrum is pure point and the eigenfunctions are exponentially localized with the localization length $\xi(E) = 1/\gamma(E)$. In contrast, for $|E| < 2$, we find $\gamma(E) = 0$. While the vanishing of the LE is generally associated to extended states and absolutely continuous spectrum, this is forbidden for the unbounded potential we consider [64]. Hence we have to conclude that the energy spectrum in the interval $[-2, 2]$ is singular continuous and the wave functions are all critical, i.e. they are neither exponentially localized nor extended, but multifractal. This mathematically proves the onset of AMEs at the energies $E_m = \pm 2$ for the quasiperiodic potential of equation (2).

# 3 Multifractality in the unbounded quasiperiodic potential

These analytic predictions are supported by numerical calculations, based on exact diagonalization of the Hamiltonian of the model (1) with the potential (2). The localization properties of a wavefunction $\psi$, normalized as $\sum_n |\psi_n|^2 = 1$, are characterized by the generalized inverse participation ratio,

$$\text{IPR}_q = \sum_n |\psi_n|^{2q} , \tag{6}$$

with $q > 1$. Since localized states are unaffected by the boundaries, they are characterized by an IPR independent of the system size $L$ (in units of the lattice spacing), *i.e.* $\text{IPR}_q \sim 1/L^{\tau_q}$ with $\tau_q = 0$. In contrast, an extended state in dimension $d$ scales as the system size, *i.e.* $\tau_q = d(q-1)$, while critical states are multifractal and characterized by the scaling exponent $\tau_q = D_q(q-1)$, where $0 < D_q < 1$ is a noninteger fractal dimension. In the latter two cases, the $\text{IPR}_q$ vanishes in the thermodynamic limit, for $q > 1$. Without loss of generality, in practical numerical calculation, we focus on the $q = 2$ case and ignore the subscript $q$. Numerical results for the IPR versus the potential amplitude $V$ and the eigenenergy $E_j$ of the $j$th eigenstate, run for a large system of $L = 1000$ sites, are shown in Fig. 1(b). Consistently with the analytical predictions, they show clear transitions between localized states (characterized by a finite IPR) for $|E| > 2$ and nonlocalized states (characterized by a vanishingly small IPR) for $|E| < 2$. Note that the phase $\theta$ in Eq. (1) is essentially irrelevant (see Appendix B).

To further characterize the wave functions, we use multifractal analysis. The size of the system $L$ is chosen as the $m$th Fibonacci number $F_m$. The advantage of this arrangement is that the inverse golden number can be approximately replaced by the ratio of two successive Fibonacci numbers, i.e., $\alpha = (\sqrt{5}-1)/2 = \lim_{m\to\infty} F_{m-1}/F_m$, see for instance ref. [40]. Then, for each wave function $\psi_n^j$, a scaling exponent $\beta_n^j$ can be extracted from the $n$th on-site probability $P_n^j = |\psi_n^j|^2 \sim (1/F_m)^{\beta_n^j}$. According to multifractal analysis, when the wave functions are extended, the maximum of $P_n^j$ over $n$ scales as $\max_n(P_n^j) \sim 1/F_m$, i.e., $\beta_{min}^j \equiv \min_n(\beta_n^j) = 1$. On the other hand, when the wave functions are localized, $P_n^j$ peaks at very few sites and is nearly zero at the other sites, yielding $\max_n(P_n^j) \sim \text{const.}$ and $\beta_{\min}^j = 0$. As for the critical wave functions, the corresponding $\beta_{\min}^j$ is located within the interval $(0, 1)$, and can be used to discriminate extended and critical states. The multifractality analysis of the parameter $\beta_{\min}^j$ applied to standard quasiperiodic models, namely the Aubry-André model and that of Eq. (2) but with $0 < a < 1$, confirms this intuitive picture (see Appendix C). As usual, significant fluctuations are oberved at criticality. To reduce these fluctuations, we use the average scaling exponent, $\beta_{min} = \frac{1}{L'} \sum \beta_{min}^j$, over the $L'$ wave functions either in the energy interval $(-2, 2)$ or outside it.

Figure 2(a) shows the scaling exponent $\beta_{\min}$ as a function of the inverse Fibonacci index $1/m$ for various amplitudes $V$ of the quasiperiodic potential. For states with an eigenenergy within the interval $(-2, 2)$ (brown markers), we find that $\beta_{\min}$ has finite values between and strictly different from 0 and 1. Extrapolating linearly to the thermodynamic limit, $1/m \to 0$, we find that $\beta_{\min}$ asymptotically tends to about 0.4, nearly independently of the potential amplitude $V$, clearly indicating the criticality of corresponding wave functions. In contrast, for states with an eigenenergy outside the interval $(2, 2)$ (red markers), $\beta_{min}$ asymptotically tends to zero in the thermodynamic limit, indicating that the corresponding wave functions are localized. This clearly confirms the onset of AMEs, separating an energy interval of critical states in the energy interval $(-2, 2)$ from localized states outside it.

Critical states appearing in the energy interval $(-2, 2)$ can also be distinguished from extended states using the scaling of the IPR. Figure 2(b) shows the mean value of the IPR, $MIPR = \frac{1}{L'} \sum IPR_n$ over the corresponding wave functions. For wave functions with an energy

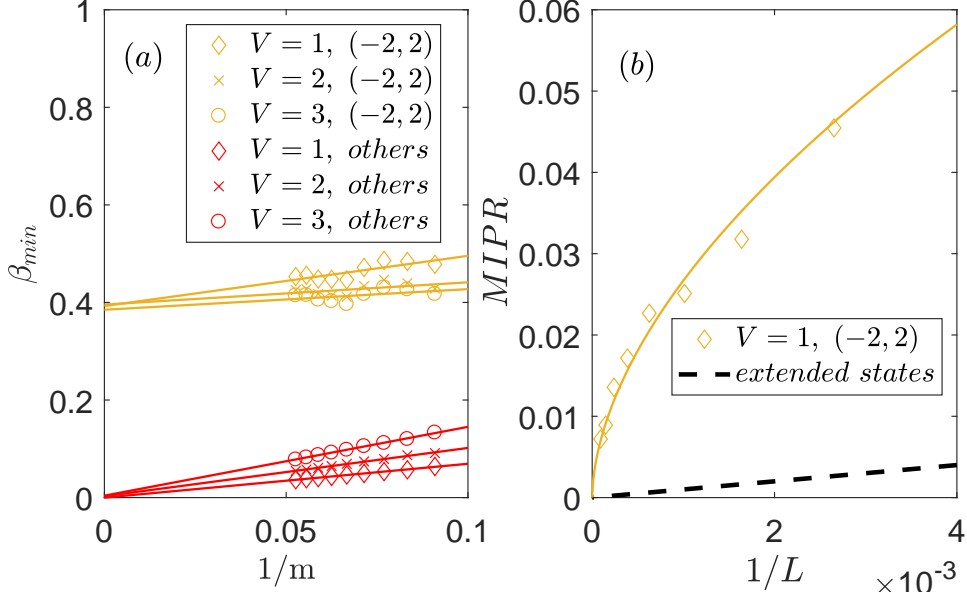

Figure 2: (a) Minimal scaling exponent $\beta_{\min}$ as a function of the inverse Fibonacci index $1/m$ for the various potential strengths $V$ and $a = 2$. The brown markers correspond to the eigenenergies in the interval $(-2, 2)$, and the red markers correspond to the other energies, i.e., outside $(-2, 2)$. (b) Mean IPR (MIPR) versus the inverse system size $1/L$ for states in the critical phase for $V = 1$ (brown markers). It shows the power law behaviour $MIPR \sim 1/L^{0.56}$, clearly different from extended states ($1/L$, shown as the dashed black line for reference).

in the interval $(-2, 2)$, we find the scaling $MIPR \sim 1/L^{0.56}$ (brown markers), clearly different from the scaling $1/L$ expected for extended sates (dashed black line). This further proves that the wave functions within the interval $(-2, 2)$ are indeed critical, rather than extended.

## 4 Quasiperiodic Su-Schrieffer-Heeger model

Having demonstrated the existence of AMEs in an exactly-solvable model, it is tempting to ask whether other models can support such AMEs. The mathematical treatment above suggests that diagonal, unbounded quasiperiodic models such that $\gamma$ vanishes on a certain energy interval are good candidates. Here, we rather consider another class of models, namely bounded models with off-diagonal quasi-periodicity. The simplest one consists in considering a tight-binding model with a hopping modulated by a quasiperiodic term, $t_n = 1 + V_n$, with $V_n = V \cos(2\pi a n + \theta)$. This model, however, displays a critical amplitude at $V = 1$ but no ME [69]. To remedy this issue, consider the quasiperiodic Su-Schrieffer-Heeger (SSH) chain governed by the eigenvalue problem

$$
\begin{aligned}
E a_n &= (1 + \lambda) b_{n-1} + (1 - \lambda + V_n) b_n, \\
E b_n &= (1 - \lambda + V_n) a_n + (1 + \lambda) a_{n+1},
\end{aligned}
\tag{7}
$$

where $a_n$ and $b_n$ are the wave function amplitudes on the sublattices A (blue spheres) and B (red spheres) of the $n$-th unit cell, respectively, see Fig. 3(a). The $b_{n-1}$-$a_n$ bond (double red bond) is the usual strong bond of hopping amplitude $t_n = 1 + \lambda$, with $\lambda$ the dimerization strength, while the weak bond $a_n$-$b_n$ (single blue bond) is modulated quasi-periodically, $t'_n = 1 - \lambda + V_n$. In the absence of an analytical solution for this model, we exclusively rely

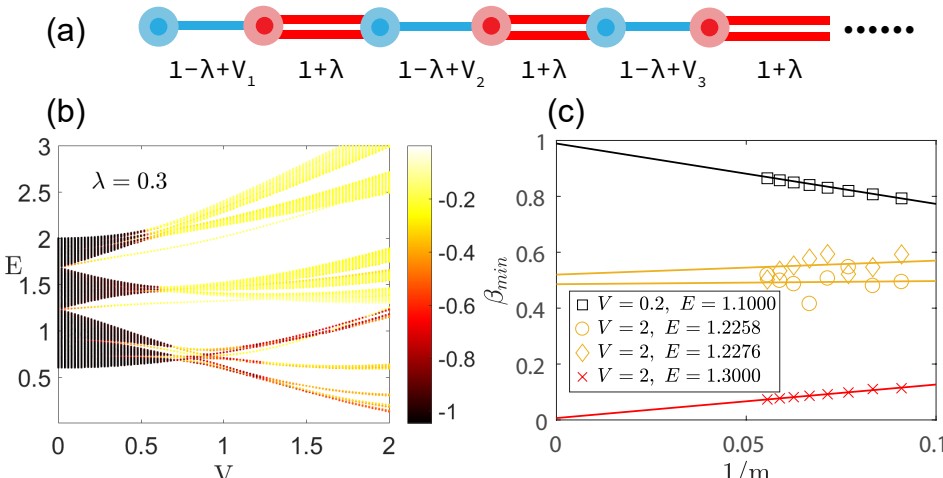

Figure 3: (a) Cartoon picture of the quasiperiodic Su-Schrieffer-Heeger chain. The blue and red spheres represent the sublattices A and B, respectively. The $b_{n-1}$-$a_n$ bond has a uniform hopping amplitude $t_n = 1 + \lambda$, with $\lambda$ the dimerization strength, the $a_n$-$b_n$ bond is modulated quasi-periodically, $t'_n = 1 - \lambda + V_n$. (b) Opposite of the IPR scaling exponent, $-\tau$, versus potential amplitude $V$ and eigen energy $E$. (c) Minimal exponent $\beta_{\min}$ of individual eigenstates in the three phases (localized, extended, and critical) as a function of the inverse Fibonacci index $1/m$ for the various potential strengths $V$ and energies $E$.

on numerics. To emphasize the difference between extended, critical, and localized states, we consider the IPR scaling exponent, $\tau = -d \log(\text{IPR})/d \log(L)$, rather than the bare value of the IPR. A typical result of $-\tau$ versus the quasiperiodic amplitude $V$ and the eigenstate energy $E$ is shown in Fig. 3(b) for the dimerization strength $\lambda = 0.3$. It yields a rich phase diagram comprising extended ($\tau \simeq 1$, black points), localized ($\tau \simeq 0$, bright yellow points), and critical ($0 < \tau < 1$, orange-red points) states [1]. Similar results are found for other values of $\lambda$ (see Appendix D). The spectrum splits into three main branches. For weak quasi-periodic modulation $V$, all states are extended up to a branch-dependent critical value. For the upper two branches, the states are localized for $V$ above the critical value. On the lower main branch, however, narrow energy intervals of critical states, characterized by $0 < \tau < 1$, appear on the lower and upper parts of the main branch, for $V \gtrsim 0.7$. Moreover, for $0.7 \lesssim V \lesssim 1.2$, the intermediate states also appear critical. This points towards the existence of one or several AMEs, depending on the value of $V$. These results are further supported by the finite-size scaling analysis of the $\beta_{\min}$ exponent of individual states picked up in the various phases, see Fig. 3(c). For extended and localized states, $\beta_{\min}$ tends towards 1 and 0, respectively, in the thermodynamic limit. In contrast, for critical states, the asymptotic limit yields $0 < \beta_{\min} < 1$.

While the exact determination of the AME in this model is still an open question, it is tempting to interpret it by analogy with the diagonal model discussed above. The results of Fig. 3(b) for $\lambda = 0.3$, as well as those obtained for other values of $\lambda$ (see Appendix D), suggest that AMEs are found for $V \gtrsim 1 - \lambda$, that is the point where some bounds have arbitrary small (although non zero) hopping terms. Hence vanishingly small but finite values of hopping amplitudes in the off-diagonal quasiperiodic SSH model play a similar role as arbitrary large but finite values of the on-site potential in the diagonal model.

---

[1] Note that due to the chiral symmetry of the quasiperiodic SSH model, we just need to plot the positive energy sector, $E > 0$. Since we do not study topological properties here, we do not plot the $E = 0$ case.

# 5 Physical implementations

We now briefly discuss physical realizations of the models considered in this work, considering first the diagonal unbounded model of Eqs. (1) and (2). The physical realization of an unbounded potential may be difficult for it involves arbitrary large energies. To avoid this issue, we may instead use Floquet engineered Hamiltonians. A periodically-kicked quantum or classical system is described rather generally by the Schrödinger equation

$$i\frac{\partial \Psi}{\partial t} = K(p)\Psi + V(x)\sum_n \delta(t-n)\Psi, \qquad (8)$$

for the wave function $\Psi = \Psi(x,t)$, where $x$ and $p$ are conjugate variables. Working along the lines of refs. [70, 71], the Floquet eigenvalue problem associated to Eq. (28) is mapped onto the spectral problem of Eqs. (1) and (2) provided the permanent and kicked components are engineered such that

$$K(p) = -2\text{atan}[aE\cos(p)],$$

and

$$V(x) = -2\text{atan}[2\cos(2\pi\alpha x)].$$

Such terms in the Hamiltonian can be emulated in various systems (see details in Appendix E). For instance, one may use propagation of light waves in lens guides or optical resonator systems [72]. Such optical systems have been exploited to observe phenomena like dynamical localization and quantum chaos (see e.g. refs. [73–77] and references therein). In order to realize the model considered here, one may use a Fabry-Perot optical cavity in a self-imaging configuration [78–81], formed by two flat end mirrors with two intracavity focusing lenses of focal length $f$ and appropriately tailored phase gratings placed at near- and far-field planes of the cavity. The eigenvalue equation that defines the optical modes of the cavity is precisely Eq. (1) with the irrational $\alpha = \lambda f/(A_1 A_2)$, where $\lambda$ is the light wavelength and $A_{1,2}$ are the spatial periods of the two gratings.

In a different physical context, one may use ultracold atoms in a bichromatic optical lattice made of a strong primary lattice and a second shallow lattice, similarly as in ref. [23]. Here we propose to periodically kick the primary lattice from a large to a weak value, hence kicking the hoping term. In contrast to previous realizations of kicked quantum rotators [82, 83], here we kick the kinetic term instead of the potential term, and localization should, correspondingly, be observed in real space instead of momentum space. Crucially, it permits to engineer the unbounded potential $v(x)$ using bounded lattices. The system is then governed by Eq. (8) with exchanged position and (quasi-)momentum, $x \to p$ and $p \to -x$. The periodicity of $V(p)$ is realized from the dispersion relation of the Bloch waves in the tight-binding regime. The bounded potential $K(x)$ can finally be appropriately designed within the secondary lattice using standard digital micromirror device (DMD) methods.

The quasiperiodic SSH model may be realized in different quantum or classical settings using matter, electromagnetic or acoustic waves, such as in dimerized lattices of Rydberg atoms [84], atomic wires with modulated hopping energies [85], arrays of optical waveguides or coupled resonators with engineered evanescent mode coupling [86, 87], and acoustic waveguide structures [88]. For example, in ref. [84] atoms in Rydberg states trapped in a controlled array of optical tweezers were used to emulate the standard SSH model. In this experiment, the alternating hopping energies were controlled by alternating the distance between the trapping sites between a large and a smaller value. Such an approach may be extended to eumulate the model of Eq. (7) by further modulating quasiperiodically the shortest distance.

## 6 Discussion

In summary, we have shown the emergence of AMEs, separating energy intervals of localized states from energy intervals of critical states, in various quasi-periodic models. On the one hand, we have rigorously demonstrated the existence of AMEs in an exactly solvable diagonal model and validated in numerical calculations using multifractal analysis. On the other hand, we have extended the concept to a quasi-periodic off-diagonal SSH model and obtained clear evidence of AMEs in numerical calculations.

These results pave the way to both experimental and theoretical developments. On the one hand, we have shown that the models proposed here can be emulated in photonic systems and ultracold atomic gases. Other platforms allowing a controlled design of various quasi-periodic structures, such as polariton condensates, could also be considered. On the other hand, while our mathematical proof suggests an approach to build unbounded quasi-periodic models hosting AMEs, our results leave open the fundamental question of understanding the necessary and sufficient conditions for a quasi-periodic model to host such AMEs. In this respect, it would be interesting to extend our results to other classes of quasi-periodic systems displaying AMEs, including either bounded or unbounded models, as well as to non-Hermitian quasi-crystals.

## 7 Acknowledgments

T. L. was supported by the Natural Science Foundation of Jiangsu Province (Grant No. BK20200737), NUPTSF (Grants No. NY220090 and No. NY220208), the National Nature Science Foundation of China (Grants No. 62071248, and No. 12074064), and the Innovation Research Project of Jiangsu Province (Grant No. JSSCBS20210521). X. X. is supported by Nankai Zhide Foundation. S. L. acknowledges the Spanish State Research Agency, through the Severo Ochoa and Maria de Maeztu Program for Centers and Units of Excellence in R&D (Grant No. MDM-2017-0711). L. S.-P. acknowledges support from GENCI-CINES (Grant No. 2020-A0070510300).

## A Lyapunov exponent analysis and anomalous mobility edges

Here we detail the analytical derivation of the Lyapunov exponent using ideas of Avila's global theory [66], suitably extended to the case of unbounded potentials [68]. The Lyapunov exponent (LE) $\gamma_0(E)$ for the spectral problem with incommensurate potential $v_n$

$$\psi_{n+1} + \psi_{n-1} + v_n \psi_n = E\psi_n, \tag{9}$$

with $v_n = v(x = 2\pi\alpha n + \theta)$, $v(x) = V/[1-a\cos(x)]$ ($a > 1$), $\alpha$ irrational, is defined as [66–68]

$$\gamma_0(E) = \lim_{n\to\infty} \frac{1}{2\pi n} \int_0^{2\pi} d\theta \log \|T_n(\theta)\|, \tag{10}$$

where $\|T_n(\theta)\|$ is the norm of the $2 \times 2$ transfer matrix $T_n(\theta)$, given by the ordered product

$$T_n(\theta) = \prod_{l=0}^{n-1} \begin{pmatrix} E - v(2\pi\alpha l + \theta) & -1 \\ 1 & 0 \end{pmatrix} = \prod_{l=0}^{n-1} T(2\pi\alpha l + \theta), \tag{11}$$

with

$$T(\theta) = \begin{pmatrix} E - v(\theta) & -1 \\ 1 & 0 \end{pmatrix} = \begin{pmatrix} E - \frac{V}{1-a\cos\theta} & -1 \\ 1 & 0 \end{pmatrix}. \tag{12}$$

Let us consider a complex extension of the LE, denoted by $\gamma_\epsilon(E)$, which is obtained from Eq. (10) by letting $\theta \to \theta + i\epsilon$, with $\theta$ and $\epsilon$ real, i.e.

$$\gamma_\epsilon(E) = \lim_{n \to \infty} \frac{1}{2\pi n} \int_0^{2\pi} d\theta \log ||T_n(\theta + i\epsilon)||. \tag{13}$$

To apply Avila's global theory [66], we remove the singularity of $T(\theta)$ by letting [68]

$$T(\theta) = \frac{1}{1 - a \cos\theta} B(\theta), \tag{14}$$

with matrix elements $B(\theta)$ analytic functions of $\theta$. One then readily obtains

$$\gamma_\epsilon(E) = \frac{1}{2\pi} \int_0^{2\pi} d\theta \log \frac{1}{|1 - a \cos(\theta + i\epsilon)|} + \gamma_\epsilon^1(E), \tag{15}$$

i.e.

$$\gamma_\epsilon(E) = -|\epsilon| - \log\left(\frac{a}{2}\right) + \gamma_\epsilon^1(E), \tag{16}$$

where we have set

$$\gamma_\epsilon^1(E) = \lim_{n \to \infty} \frac{1}{2\pi n} \int_0^{2\pi} d\theta \log ||B_n(\theta + i\epsilon)||. \tag{17}$$

Since $B$ is an analytic function of $\theta$, it follows that $\gamma_\epsilon^1(E)$ is a continuous function of $\epsilon$. Hence $\gamma_\epsilon(E)$ is a continuous function of $\epsilon$ as well because of Eq. (16). To calculate $\gamma_\epsilon(E)$ at $\epsilon = 0$, we can thus compute the limit of $\gamma_\epsilon(E)$ as $|\epsilon| \to 0$. To compute $\gamma_\epsilon(E)$ for $\epsilon > 0$, let us first consider the limit $\epsilon \to \infty$. Uniformly in $\theta$, one has

$$T(\theta + i\epsilon) = T_\infty \left[1 + O(\exp(-\epsilon))\right], \tag{18}$$

where

$$T_\infty = \begin{pmatrix} E & -1 \\ 1 & 0 \end{pmatrix}, \tag{19}$$

so that one readily obtains

$$\begin{aligned} \gamma_\epsilon(E) &= \lim_{n \to \infty} \frac{1}{n} \log \left\| \begin{pmatrix} E & -1 \\ 1 & 0 \end{pmatrix}^n \right\| + O(1) \\ &= \log \left| \frac{E \pm \sqrt{E^2 - 4}}{2} \right| + O(1), \end{aligned} \tag{20}$$

as $\epsilon \to \infty$. In Eq. (20), the $\pm$ sign on the right hand side should be chosen so that to get the largest value of $\gamma_\epsilon$. Since for $\epsilon \neq 0$ $(A, \alpha)$ is an analytic cocycle, we can apply the quantization theorem of acceleration [66], so that

$$\gamma_\epsilon(E) = \log \left| \frac{E \pm \sqrt{E^2 - 4}}{2} \right|, \tag{21}$$

for all $\epsilon$ sufficiently large. In addition, due to the convexity, continuity and symmetry of $\gamma_\epsilon(E)$ (i.e. $\gamma_{-\epsilon}(E) = \gamma_\epsilon(E)$), one necessarily has

$$\gamma_\epsilon(E) = \log \left| \frac{E \pm \sqrt{E^2 - 4}}{2} \right|, \tag{22}$$

for any $\epsilon$ (including $\epsilon = 0$ for continuity), i.e. the Lyapunov exponent $\gamma_\epsilon(E)$ is independent of $\epsilon$. Such a property is analogous to the behavior found in the Maryland model [68] and

closely related to the unbounded nature of $v(x)$. However, as compared to the Maryland model, in our model the Lyapunov exponent $\gamma_0(E)$ is not always strictly positive. In fact, we have $\gamma_0(E) > 0$ only for $|E| > 2$. In this region, we conjecture that, like for the Maryland model [50, 68], for $\alpha$ irrational Diophantine the spectrum is pure point with exponentially-localized eigenfunctions and localization length given by $\xi(E) = 1/\gamma_0(E) = 1/\ln|\frac{E \pm \sqrt{E^2-4}}{2}|$. This result is confirmed by the numerical analysis given in the main text and in Appendix C. In contrast, for $|E| < 2$ one has $\gamma_0(E) = 0$. Since for unbounded potentials the absolutely continuous spectrum is empty [64], we conclude that the energy spectrum in the interval $(-2, 2)$ is singular continuous, and corresponding wave functions are critical, i.e. they are not exponentially localized neither extended in the Bloch's sense (see Appendix C). Remarkably, the mobility edges are independent of potential parameters $V$ and $a$ (with $|a| > 1$).

## B    Influence of the phase $\theta$

Figure 4 shows that the phase $\theta$ in Eq. (1) does not affect the spectrum nor the localization properties of the model (2).

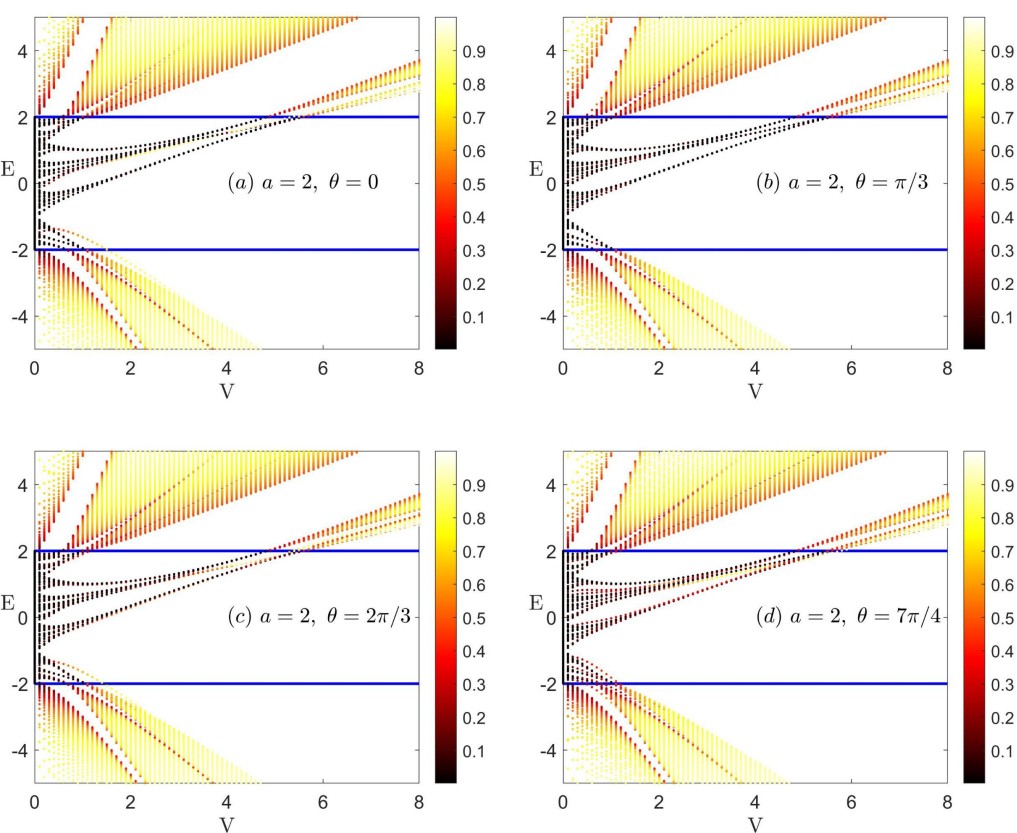

Figure 4:  Inverse participation ratio IPR versus potential amplitude and energy for the model of Eqs. (1) and (2), and various values of the phase $\theta$.

# C Multifractal analysis of quasiperiodic models and spatial distributions of wave functions

Here we provide numerical results for the multifractal analysis of various quasiperiodic models.

Let us start with the same model as considered in the main paper but in the bounded case [53], $0 < a < 1$, see Fig. 5(a). In this case, there exists a standard mobility edge $E_m$ separating localized from extended states. Correspondingly, we find that, for extended states ($E < E_m$, black markers), $\beta_{\min}$ tends to 1 in the thermodynamic limit. In contrast, for localized states ($E > E_m$, red markers), $\beta_{\min}$ tends 0. In both cases, a clear linear behaviour of $\beta_{\min}$ versus the inverse Fibonacci index $1/m$ is found.

In contrast, the Aubry-André model does not display any mobility edge but a criticall potential amplitude at $V = 2$. In the extended pahse, $V < 2$, we find that $\beta_{\min}$ tends to 1 in the thermodynamic limit, see Fig. 5(b) (black makers). When, instead, $V > 2$, the system is in the localized phase and the corresponding $\beta_{\min}$ tends to 0 (red markers). At the phase transition, $V = 2$, the system is in the critical phase and the corresponding $\beta_{\min}$ is clearly within the interval $(0, 1)$ (brown markers). Significant fluctuations are observed as a function of the Fibonacci index $m$. However, these states are clearly be distinguished from extended and localized states. This proves that the multifractal analysis can be used to distinguish extended, critical, and localized states in a wide variety of quasiperiodic systems.

We have also implemented the numerical calculation for the unbounded model discussed in the main paper. The corresponding results are shown in the main text for the tuning parameter $a = 2$ and in Fig. 5(c) and (d) for $a = 6$ and $a = 11$. They confirm the existence of AMEs with all states critical in the energy range $(-2, 2)$ and all states localized otherwise.

The onset of AMEs can be also explicitly confirmed by an inspection of the spatial distributions of wave functions. Figure 6 plots the wave functions of six eigenenergies separated by the AME $E_m = 2$ when $V = 2$. An inspection of the figure clearly indicates that the wave functions with eigenenergies above $E_m$ [panels (b), (d), and (f)] are maximally localized at one site of the chain. In contrast, the wave functions with corresponding eigenenergies below

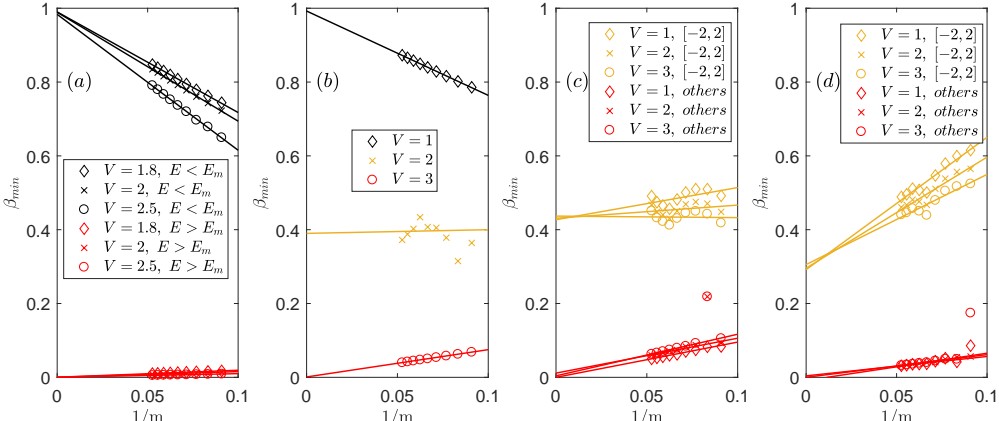

Figure 5: (a) $\beta_{min}$ as a function of the inverse Fibonacci index $1/m$ for the model of Ref. [53] (same as our model but with $0 < a < 1$) and $a = 0.5$. The ME is $E_m = 2(2 - V) = 0.4$ for $V = 1.8$, $E_m = 0$ for $V = 2$, and $E_m = -1$ for $V = 2.5$. (b) $\beta_{min}$ as a function of the inverse Fibonacci index $1/m$ for the Aubry-André model. (c) $\beta_{min}$ as a function of the inverse Fibonacci index $1/m$ for the model in this paper with the tuning parameter $a = 6$. (d) $\beta_{min}$ as a function of the inverse Fibonacci index $1/m$ for the model in this paper with the tuning parameter $a = 11$.

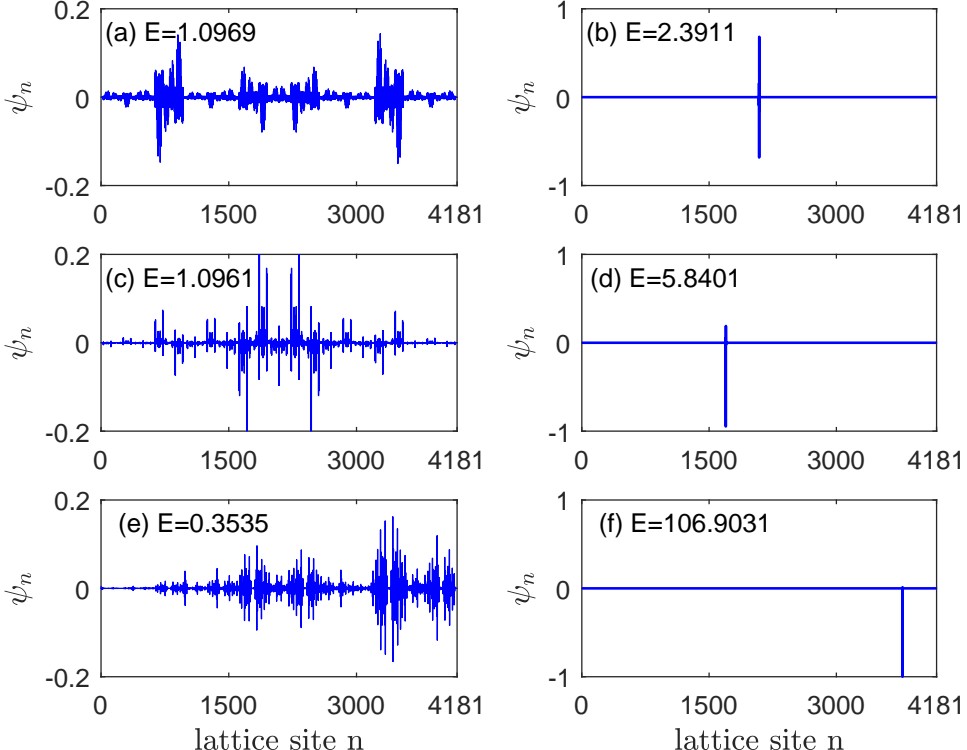

Figure 6: Spatial distributions of $\psi_n$ for a few eigenfunctions with eigenenergies either below or above the AME $E_m = 2$. A lattice with $L = 4181$ sites has been used in numerical simulations. Here we choose six eigenenergies (with four significant digits): critical states below $E_m$ [(a), (c) and (e)], and localized states above $E_m$ [(b), (d) and (f)].

$E_m$ [panels (a), (c), and (e)] are neither localized nor extended over the whole space. Instead, they display clear self-similarities, which is the characteristic of critical states. This confirms that the AME $E_m = 2$ distinctly separates localized from critical states.

# D  Additional results for the SSH model

Here we give additional results for the inverse participation ratio of the SSH model. Figure 7 shows the counterpart of Fig. 3(b) of the main paper for two other values of the dimerization parameter, $\lambda = 0$ [Figure 7(a)] and $\lambda = 0.5$ [Figure 7(b)].

The spectrum splits into three main branches, clearly identifiable at $V \simeq 0$. For low values of $V$, all states are extended (black points). For the two upper branches, we find a transition to localized states (yellow dots). For the lower branch, however, we obtain energy intervals where the states are critical (red-orange dots). The critical value of the potential is compatible with the simple estimate $V \simeq 1 - \lambda$ [see also Fig. 3(b) of the main paper]. Beyond the critical point, we obtain AMEs separating critical energy intervals from localized energy intervals.

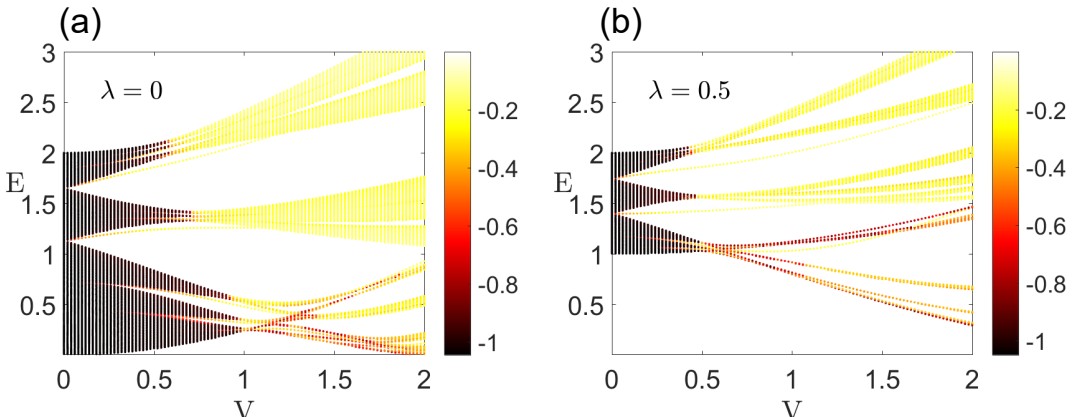

Figure 7: Opposite of the IPR scaling exponent, $-\tau = \log(\text{IPR})/\log(L)$, versus potential amplitude $V$ and eigen energy $E$ for the quasiperiodic SSH model of Eq. (7) in the main paper, with (a) $\lambda = 0$ and (b) $\lambda = 0.5$.

## E Physical implementations of the unbounded potential model

### Spectral problem of a periodically-kicked system

Spectral problems involving diverging potentials $v(x)$ on a lattice are known to arise in periodically-kicked classical or quantum systems, such as in the periodically-kicked quantum rotator model [70, 71] or in its linear version, known as the Maryland model [48–50, 89, 90], and they are related to major physical effects such as dynamical and Anderson localization. In the kicked rotator model, the kinetic energy $K(p)$ is a quadratic function of the momentum $p$, while in the Maryland model $K(p)$ is assumed to be linear in $p$. In the latter case, the unbounded potentials $v(x)$ is described by the trigonometric tangent function. The ability to engineer the kinetic energy $K$ and the potential term $V$ in the Schrödinger equation can give rise to spectral problems on the lattice with different and tailored unbounded potentials $v(x)$.

Let us consider rather generally the dynamics of a one-dimensional periodically-kicked quantum particle, described by the dimensionless Schrödinger equation

$$i\frac{\partial \Psi}{\partial t} = K(\hat{p})\Psi + V(x)\sum_n \delta(t - n)\Psi, \tag{23}$$

for the wave function $\Psi = \Psi(x, t)$, where $\hat{p} = -i\partial_x$, $K(p)$ is the dispersion relation of the kinetic energy term, and $V(x)$ is the external potential. The evolution of the wave function before each kick, $\Psi^{(m)}(x) = \Psi(x, t = m^-)$, is governed by the following map

$$\Psi^{(m+1)}(x) = \exp[-iK(\hat{p}_x)]\exp[-iV(x)]\Psi^{(m)}(x). \tag{24}$$

After setting $\Psi^{(m)}(x) = \Psi(x)\exp(-i\mu m)$, where $\mu$ is the Floquet quasi-energy which varies in the range $(-\pi, \pi)$, the following spectral problem is obtained

$$\exp(-i\mu)\Psi(x) = \exp[-iK(\hat{p})]\exp[-iV(x)]\Psi(x). \tag{25}$$

Following the method outlined in Refs. [70, 71], let us introduce the auxiliary potential $W(x) = \tan[V(x)/2]$ so that

$$\exp[-iV(x)] = \frac{1 - iW(x)}{1 + iW(x)}. \tag{26}$$

After setting

$$\psi(x) = \frac{\Psi(x)}{1 + iW(x)}, \tag{27}$$

from Eqs. (25), (26), and (27), one obtains

$$[1 + iW(x)]\psi(x) = \exp[i\mu - iK(-i\partial_x)]\{[1 - iW(x)]\psi(x)\}. \tag{28}$$

Let us now assume that the potential $V(x)$, and thus the function $W(x)$, is a periodic function of $x$ with period $1/\alpha$, so that $W(x) = \sum_n W_n \exp(2\pi i \alpha n x)$, where $\alpha$ is irrational. We can thus search for a solution to Eq. (28) of the Bloch form, $\psi(x) = \sum_n \psi_n \exp(2\pi i \alpha x n + i\theta x)$, with $\theta$ constant. From Eq. (28), it follows that the Fourier coefficients $\psi_n$ satisfy the equation

$$\psi_n + i \sum_l W_{n-l}\psi_l = \{\exp[i\mu - iK(2\pi\alpha n + \theta)]\} \left( \psi_n - i \sum_l W_{n-l}\psi_l \right), \tag{29}$$

Equation (29) is solved by letting

$$\sum_l W_{n-l}\psi_l = S_n \psi_n, \tag{30}$$

with

$$\frac{1 + iS_n}{1 - iS_n} = \exp[i\mu - iK(2\pi\alpha n + \theta)]. \tag{31}$$

To obtain a tight-binding model with nearest-neighbor hopping, let us assume a potential $V(x)$ such that $W(x) = -2\cos(2\pi\alpha x)$, i.e.

$$V(x) = -2\operatorname{atan}\left[2\cos(2\pi\alpha x)\right]. \tag{32}$$

In this case, from Eqs. (30), (31), and (32), one finally obtains

$$\psi_{n+1} + \psi_{n-1} + v(2\pi\alpha n + \theta)\psi_n = E\psi_n, \tag{33}$$

where we have set

$$E \equiv \frac{1}{\tan(\mu/2)}, \tag{34}$$

and

$$v(x) = \frac{1 + E^2}{E} \frac{1}{1 + \frac{1}{E}\tan\left(\frac{K(x)}{2}\right)}. \tag{35}$$

Clearly, Eq. (33) describes the spectral problem of a tight-binding lattice with a potential $v(x)$, which depends on the energy $E$. However, such a dependence of the potential on the energy is not a major issue for the model discussed in our work and for the appearance of anomalous mobility edges. In fact, let us assume a periodic kinetic energy $K(p)$ of the form

$$K(p) = -2\operatorname{atan}\left[D\cos(p_x)\right]. \tag{36}$$

The corresponding potential reads as

$$v(x) = \frac{V}{1 - a\cos(x)}, \tag{37}$$

with $V = (1 + E^2)/E$ and $a = D/E$, i.e. the model considered in the main text. Since the spectral properties and mobility edges for the potential given by Eq. (37) are independent of the potential amplitude $V$ and the only condition for an unbounded potential is $|a| > 1$, we

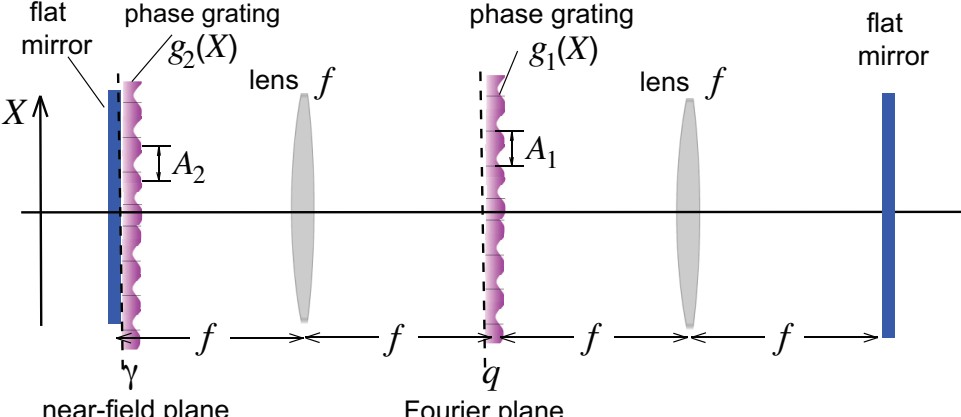

Figure 8: Schematic of a self-imaging optical resonator with flat end mirrors and with two phase gratings placed at the near-field (flat end mirror at the left side) and Fourier (far-field) planes. The spatial periods of the two gratings are $A_1$ and $A_2$, respectively. The irrational $\alpha$ is defined in terms of physical parameters by $\alpha = \lambda f / (A_1 A_2)$, where $f$ is the focal length and $\lambda$ is the light wavelength.

can conclude that critical mobility edges, separating exponentially localized states and critical states for irrational $\alpha$ with Diophantine properties, arise at the energy $E = E_m = 2$, i.e. at the quasi-energy $\mu_m = 2\,\mathrm{atan}(1/2)$, whenever the condition $D > 2$ is satisfied.

In the previous analysis we assumed that the quantum particle is periodically kicked by an external potential, however likewise one could kick the kinetic energy term instead of the potential term. In the latter case localization should be observed in real space instead of momentum space.

We now suggest two possible physical implementations of periodically-kicked systems, a classical system (the optical resonator mode) and a quantum system (ultracold atoms in a kicked bichromatic optical lattice).

**The optical resonator model**

Here, we show that the map (24) and the associated spectral problem (25) naturally arise in the calculation of cavity modes of light waves in an optical resonator [72]. We note that wave and ray propagation of light in lens guides and optical resonator systems have been often employed to study and observe phenomena like dynamical localization and quantum chaos (see e.g. refs. [73–77] and references therein). Specifically, let us consider a Fabry-Perot optical cavity in so-called self-imaging (or 4-$f$) configuration [78–81], formed by two flat end mirrors with two focusing lenses of focal length $f$, as schematically shown in Fig. 8. For the sake of simplicity, we assume a one-transverse spatial dimension $X$. Two phase gratings, with transmission field amplitudes $t_2(X) = \exp[-ig_2(X)/2]$ and $t_1(X) = \exp[-ig_1(X)/2]$, are placed at the near-field and far-field planes ($\gamma$ and $q$) of the resonator, as shown in Fig. 8. The spatial period of the two gratings are $A_1$ and $A_2$, respectively, i.e. $g_1(X + A_1) = g_1(X)$ and $g_2(X + A_2) = g_2(X)$. In the scalar and paraxial approximations, wave propagation at successive transits inside the optical cavity can be readily obtained from the generalized Huygens integral by standard methods [72]. Neglecting finite aperture effects, the field envelope $\Psi^{(m)}(X)$ of the progressive wave at the reference plane $\gamma$ in the cavity and at the $m$-th round-trip evolves according to the map

$$\Psi^{(m+1)}(X) = \exp[-ig_2(X)] \int_{-\infty}^{\infty} d\theta \, Q(X, \theta)\Psi^{(m)}(\theta), \qquad (38)$$

where the kernel $Q$ of the integral transformation is given by [80, 81]

$$Q(X, \theta) = \left( \frac{1}{\lambda f} \right) \int d\xi \exp \left[ -i g_1(\xi) + \frac{2\pi i \xi (X - \theta)}{\lambda f} \right], \tag{39}$$

and $\lambda = 2\pi/k$ is the optical wavelength. In writing Eq. (39), we assumed $g_1(-X) = g_1(X)$. Taking into account that the integral transformation $Q$ can be written as the exponential of a differential operator, namely

$$\int_{-\infty}^{\infty} d\theta \, Q(X, \theta) \Psi(\theta) = \exp \left[ -i g_1 \left( -i \frac{\lambda f}{2\pi} \frac{\partial}{\partial X} \right) \right] \Psi(X), \tag{40}$$

after introduction of the dimensionless spatial variable $x \equiv A_1 X/(\lambda f)$, the map Eq. (38) can be written in the form of Eq. (24) with kinetic energy and potential terms given by

$$K(p) = g_1 \left( \frac{A_1}{2\pi} p \right), \; V(x) = g_2 \left( \frac{\lambda f}{A_1} x \right). \tag{41}$$

Note that the arithmetic number $\alpha$, i.e. the frequency of $V(x)$, is defined in terms of the physical parameters $A_1, A_2, \lambda$ and $f$ (grating periods, light wavelength, and focal length) by the relation

$$\alpha = \frac{\lambda f}{A_1 A_2}. \tag{42}$$

Therefore, the profiles of $K(p)$ and $V(x)$ required to simulate the unbounded potential $v(x) = V/[1 - a \cos(x)]$ and given by Eqs. (32) and (36), are basically obtained by suitably tailoring the phase grating profiles $g_{1,2}(X)$ according to Eq. (41).

**Ultracold atoms in a kicked bichromatic optical lattice**

The proposed model may alternatively be realized in ultracold-atom systems. Consider atoms subjected to a kicked (primary) optical lattice in the tight-binding regime plus a weak (secondary) potential, see Fig. 9. The Hamiltonian of the system reads as

$$\hat{\mathcal{H}} = \hat{\mathcal{H}}_1 \sum_i \delta(t - iT) + \hat{\mathcal{H}}_2, \tag{43}$$

where $\hat{\mathcal{H}}_1 = -J\tau \sum_n \left( \hat{a}_{n+1}^\dagger \hat{a}_n + \hat{a}_{n-1}^\dagger \hat{a}_n \right)$ is the primary lattice Hamiltonian mutiplied by the duration of a kick, assumed to be much shorter than the inter-kick time $\tau \ll T$. It describes atoms tunneling from site $n$ to the nearest-neighbor sites $n \pm 1$ with the hopping energy $J$, where $\hat{a}_n$ and $\hat{a}_n^\dagger$ are, respectively, the annihilation and creation operators of an atom at site $n$. The second term, $\hat{\mathcal{H}}_2 = \sum_n \mathcal{V}_n \hat{a}_n^\dagger \hat{a}_n$, accounts for the light-shift potential induced by the secondary lattice, which modulates the energy $\mathcal{V}_n$ of an atom at site $n$. In practice, one realizes a primary optical lattice with a large amplitude, so that the tunneling is negligible, and periodically quenches this amplitude to a weaker value so that the tunneling energy acquires a finite value $J$ for a short time $\tau$. The model is realized provided the latter is much shorter than the relevant time scales, $\tau \ll T, \hbar/J$. A secondary lattice, with a much weaker amplitude, realizes the on-site energy modulation $\mathcal{V}_n$ of an atom at the site $n$ of the primary lattice. Note that the effective on-site energy changes during the kicks but this is irrelevant since the effect of $\hat{\mathcal{H}}_2$ is negligible for vanishingly short kicks.

Let us denote $|n\rangle$ the Wannier state at site $n$ in the lowest-energy band of the primary lattice, and $|q\rangle$ the corresponding Bloch states. The primary lattice Hamiltonian is diagonal in

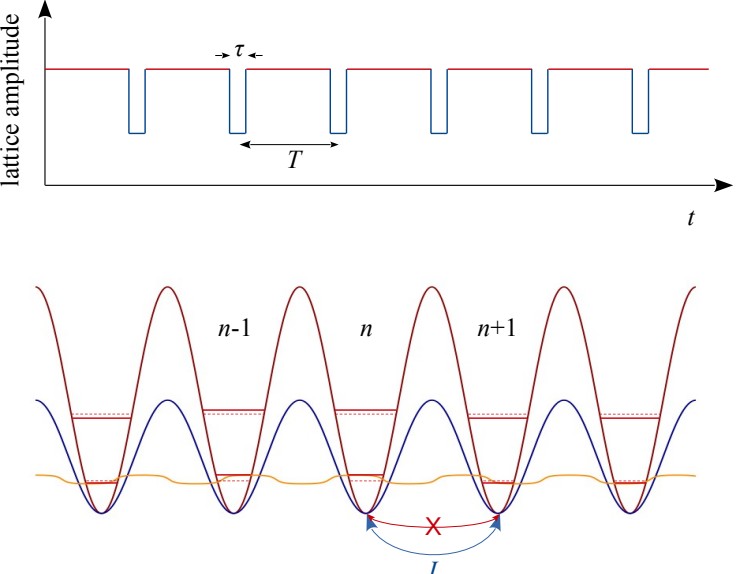

Figure 9: Ultracold-atom scheme to realize the proposed model. Lower panel: The atoms are trapped in a deep optical lattice (red line) with lattice sites labelled by $n \in \mathbb{N}$ and vanishingly small hopping amplitude. The atoms occupy the corresponding Wannier fuctions, with a uniform on-site energy (lower dotted red bars) An additional weak optical potential (orange line) modulates on on-site energies (solid red bars) according to Eq. (48). The primary lattice is periodically kicked every time $T$ to a smaller amplitude (solid blue line) for a short time $\tau$, hence setting up a finite hopping energy $J$. Upper panel: Time sequence of the lattice amplitude with color scheme consistent with the lower panel

the Bloch basis with energies $\mathcal{E}_q = -2J\cos(q)$. The single-kick evolution operator then reads as

$$\hat{U}_1 = \sum_q e^{-i\mathcal{E}_q\tau/\hbar}|q\rangle\langle q| = \frac{1 - iW(q)}{1 + iW(q)}|q\rangle\langle q|, \tag{44}$$

with

$$W(q) \simeq \frac{\mathcal{E}_q\tau}{2\hbar} = -\frac{J\tau}{\hbar}\cos(q), \tag{45}$$

since $J\tau/\hbar \ll 1$. It generates the cosine modulation of the function $W$ discussed above and the effective nearest-neighbour tight-binding hopping term $\frac{J\tau}{2\hbar}(\psi_{n+1} + \psi_{n-1})$, similar to that of Eq. (33) up to trivial rescalings of the length, $2\pi\alpha x \to q$, and energy $1 \to J\tau/2\hbar$.

The secondary lattice Hamiltonian generates the on-site term

$$S_n = \frac{J\tau}{2\hbar}\left[\frac{1 + E^2}{E}\frac{1}{1 + \frac{\tan(\mathcal{V}_n T/2\hbar)}{E}} - E\right], \tag{46}$$

with

$$E = \frac{1}{\tan(\mu T/2\hbar)}, \tag{47}$$

and $\mu$ the Floquet quasienergy. The model (33) is thus realized by setting

$$\mathcal{V}_n = -\frac{2\hbar}{T}\text{atan}\left[aE\cos(2\pi\alpha n + \theta)\right], \tag{48}$$

see Eq. (36). Such a potential varies smoothly on the primary lattice length scale, which is nothing but the optical wavelength. It can be engineered using standard digital micromirror device (DMD) techniques, which are now routinely used in ultracold-atom experiments.

It is finally worth commenting our scheme. The tight-binding model of Eq. (33) may in principle be realized directly in a static (nonkicked) primary optical lattice modulated by a secondary lattice such that $\mathcal{V}_n = \frac{V}{1 - a\cos(2\pi\alpha n + \theta)}$. This potential is, however, unbounded and cannot be striclty engineered with DMD techniques. In contrast, Floquet engineering allows one to emulate the effective unbounded potential $V_n$ using a bounded secondary optical field $\mathcal{V}_n$, which, in turn, can be realized by DMD techniques.

Kicked quantum rotator models have been previously realized with ultracold-atom systems to study dynamical localization in one and higher dimensions [82, 83]. In these works, the kicked term was a cosine potential as realized by an optical lattice periodically switched on for a short time $\tau$, and the static term was the free-particle kinetic energy. The former generated the nearest-neighbour tight-binding term of the effective model while the latter generated a quasi-disordered potential term. The latter is, however, hard to engineer beyond the quadratic (free-particle) or cosine-like (lattice-particle) dispersion relation. Our proposed implementation overcomes this issue by, instead, using the cosine dispersion relation of tightly-bound particles to generate the effective tight-binding term and the easily-engineered potential term to generate the unbounded potential of the effective model. As a result, localization is to be observed directly in real space while it was observed in momentum space in kicked rotators realized so far.

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
