# Peer review of "Anomalous mobility edges in one-dimensional quasiperiodic models"

_SciPost Physics, doi:SciPost Phys. 12, 027 (2022)_

## Round 1 · Referee Report · Anonymous (Referee 2) · 2021-10-12

Strengths

  1. The manuscript provides a thorough analysis of anomalous mobility edges (AME) in lattice models with unbounded quasi-periodic potentials. Through extensive numerics, the authors show that these potentials turn the energy states of the free energy band [-2J,2J] from delocalized to critical (multifractal), while adding new localized states outside such band.

  2. The authors show the existence of AME in a modified SSH model with quasi-periodic tunneling rates.

  3. Implementation of the studied model with cold atoms or photonic lattices is discussed.

Weaknesses

I can't see.

Report

The manuscript present interesting new results on the physics of anomalous mobility edges (AME), which are energies separating localized from critical (multifractal) states in the context of unbounded quasi-periodic lattices. The obtained results are quite interesting and I can recommend publication in this journal. Here are some points that the authors could consider to improve the presentation:

  1. In the finite-size scaling analysis for the exponent beta (Fig.2a, Fig.3c, Fig.5) the authors observed a linear dependence of beta when plotted as a function of 1/m. Since the system size L=Fm, where Fm is the m-th Fibonacci number, one finds m \propto log(L) for large enough m. The finite-size correction to the exponent beta is therefore of order 1/log(L). Is this behavior general and expected for second order phase transitions or is it specific to the model considered ?

  2. A comment to Fig.4. To my understanding, the results shown do not depend on the value of the phase because the irrational parameter alpha used in the calculation is approximated by the ratio of consecutive Fibonacci numbers. This, together with the periodic boundary conditions, makes the choice of the phase irrelevant and therefore disorder averaging can be avoided.

  3. From the Appendix A it seems that, if the disorder potential v(x) is unbounded and quasi-periodic, the Lyapunov exponent is independent of the specific form of v(x), as the final result coincides with the clean limit case. Is this statement correct?

Requested changes

  1. In Fig5: which value of a is used and what is the corresponding value of Em

  2. The authors mentioned a "multifractal theorem". What are they referring to exactly ?

  • validity: top
  • significance: high
  • originality: high
  • clarity: top
  • formatting: perfect
  • grammar: perfect

Author:  Laurent Sanchez-Palencia  on 2021-11-23  [id 1966]

(in reply to Report 1 on 2021-10-12)
Category:
answer to question

Dear Editor,

We thank you for forwarding the Referees' reports. We are grateful to both of them for the positive assessment of our work and for the comments they made. Hereafter we respond the comments from the reports and indicate the corresponding changes whenever applicable. We hope that with the changes, the paper is now suitable for publication in SciPost Physics.

Best regards, Tong Liu, Xu Xia, Stefano Longhi, and Laurent Sanchez-Palencia

Referee's comment/question 1 : "In the finite-size scaling analysis for the exponent beta (Fig.2a, Fig.3c, Fig.5) the authors observed a linear dependence of beta when plotted as a function of 1/m. Since the system size L=Fm, where Fm is the m-th Fibonacci number, one finds m \propto log(L) for large enough m. The finite-size correction to the exponent beta is therefore of order 1/log(L). Is this behavior general and expected for second order phase transitions or is it specific to the model considered ? "

Answer : The linear finite-scaling with the Fibonacci index is quite standard in quasiperiodic models. It holds for other models as well, including the celebrated Aubry-Andre-Harper (quasi Mathieu) Hamiltonian. However, we are not aware of studies establishing a direct relation between the 1/log(L) scaling of the corrections and the phase transition class.

Referee's comment/question 2 : "A comment to Fig.4. To my understanding, the results shown do not depend on the value of the phase because the irrational parameter alpha used in the calculation is approximated by the ratio of consecutive Fibonacci numbers. This, together with the periodic boundary conditions, makes the choice of the phase irrelevant and therefore disorder averaging can be avoided."

Answer : The argument suggested by the Referee is in principle correct and works in case of pure point or critical spectrum (as in our case). However, as a matter of fact in our simulations the energy spectra shown in Fig.4, for different and some illustrative values of the phase \theta, are obtained assuming a lattice size L=1000 and using open boundary conditions. The lattice size is thus not equal to nor proportional to any Fibonacci number and we directly use the irrational Diophantine number \alpha=(\sqrt{5}-1)/2. Yet, we could not observe any marked dependence of the spectra on \theta. This may be attributed to the irrational Diophantine nature of \alpha: this is why in the numerical computation of the eigenvalues of the matrix Hamiltonian for large L and rational approximant \alpha=p_n/q_n with q_n>L one cannot notice any marked sensitivity of the energy spectrum on \theta. Conversely, if one assume a system size L much larger than q_n, one could observe in the numerical simulations a dependence on \theta since the system becomes closer to periodic rather than aperiodic.

Referee's comment/question 3 : "From the Appendix A it seems that, if the disorder potential v(x) is unbounded and quasi-periodic, the Lyapunov exponent is independent of the specific form of v(x), as the final result coincides with the clean limit case. Is this statement correct?"

Answer : In fact, for a quasi-periodic and unbounded potential, the Lyapunov exponent depends rather generally on the particular form of v(x) and on the energy. For example, for the Maryland model, v(x)=V tan (\pi \alpha n + \theta), which is quasi-periodic and unbounded potential, the Lypaunov exponent L(E) can be computed analytically (see e.g. Ref.[50] in our manuscript) and the result is different than for our model. The main point is that the Lyapunov exponent in the "clean limit model" is the value of L(E) when the complex phase \epsilon added to the argument of the potential goes to infinity, while the Lypaunov exponent L(E) of the physical model, which we are looking for, is the value of L(E) when for $\epsilon=0$. Avila's global theory allows us to connect the two values of L(E), from \epsilon=\infty to \epsilon=0. However in order to use such a theory some strict conditions on the cocycle should be met.

Requested change 1 : "In Fig5: which value of a is used and what is the corresponding value of Em".

Answer : The have added the values of the tuning parameter a and the corresponding mobility edges in the caption of the figure.

  1. The authors mentioned a "multifractal theorem". What are they referring to exactly ?

Answer : We thank the referee for raising this point. Indeed, this is an incorrect wording. There is no theorem in the mathematical sense. We have changed it to "multifractal analysis", which refers to the analysis of the scalings of the parameters \beta.

---

## Round 1 · Referee Report · Anonymous (Referee 1) · 2021-10-27

Report

I am happy with the changes the authors made and would recommend the manuscript be published as it is. Regarding Fig. 4, I understand that in a localized phase, disorder average with respect to the global phase of potential usually would not lead to qualitatively change. However, in an extensive or thermalized phase, some observables (for example the level spacing statistics) are sensitive to the potential phase, and only the average value gives the meaningful prediction. In a critical phase, however, to the best of my knowledge, this is still open. I believe Fig. 4 nicely show that IPR is not sensitive to the potential phase even in a critical phase.

Requested changes

No further changes are required.

---

## Round 1 · Author Response

Dear Editor,

We thank your for forwarding the report of the Referee. We are grateful to the latter for his/her very positive report on our work. The Referee has made a couple of suggestions, which we answer below We also briefly indicate the corresponding changes we made on the manuscript and resubmit a revised version of the manuscript.

Referee's comment 1 : "The authors claim that, in a sentence below Eq. (1), that the phase "theta" is irrelevant. This is, however, usually not entirely true in the investigation of localization physics in quasiperiodic models. One usually need to take the average of "\theta" to mimic the randomness of a genetic disorder. Although this average is usually more important in the extensive phase than the localized phase. I however wonder has this \theta dependency has been considered and investigated. I can see in Eq. (3) that the analytical analysis to some degree includes this average by the integration over \theta, so the question is more concerning on the numerical analysis."

Answer : It is known from the extensive mathematical literature on the quasi-Mathieu operator (Aubry-André model) that the spectrum and localization properties of the model do not depend on the phase \theta when the incommensurate ratio \alpha is Diophantine. In our case, we use the inverse golden ratio, which is indeed a Diophantine number. Therefore, we believe that the phase is essentially irrelevant, excluding the zero-measure set of values of \theta leading to a diverging potential.

Nevertheless, we agree with the Referee that --to the best of our knowledge-- there is no rigorous proof of this conjecture for our model. To clarify this point, we have run new calculations with different values of \theta. We indeed found that the spectrum and the localization properties are unchanged, see new appendix B in the resubmitted manuscript.

Referee's comment 2 : "The energy gap is usually an important observable for the studies of localization. The gap ratio usually can give us information on whether the eigenenergies follows Gaussian Orthogonal Ensemble distribution or Poisson distribution. This is an important signature of an extensive/localized spectrum. For completeness, I suggest the authors carry out this analysis."

Answer :
The level spacing statistics is indeed a rather popular approach to determine the localization properties of disordered or quasi-periodic systems. In fact, there exists a variety of methods to determine the localization properties. In our work, we already use three of them: (i) Exact calculation of the Lyapunov exponent (for the diagonal model), (ii) scaling of the IPR, and (iii) multifractal analysis. Although we agree with the Referee that a study of the level spacing statistics could be performed, we think it goes beyond the scope of our study and it would not provide us with additional information about the onset of anomalous mobility edges in our models.

Referee's comment 3 : "This is a very minor problem, but I am a bit confused about the choice of their word "band". In the introduction, the third paragraph, the authors claim that "... can give rise to a full band of critical states ...". In condensed matter physics, a "band" usually refers to a set of eigenenergies that are close to each other, with a relatively large gap between the different bands. In Figure 1(b), one can see that the spectrum consists of multiple bands, and these bands can sometimes cross the mobility edge. Therefore, I think the term "band" might be a bit misleading. That being said, this seems to me only a problem of choice of word."

We thank the Referee for pointing out this possible confusing choice of a word. In the revised version of the manuscript, we now use the more neutral word "energy interval" instead of "band".

---

## Round 1 · List of Changes

• "bands" changes into "energy intervals"

  • Discussion about the effect of the phase enhanced and shifted to end of paragraph below Eq. (6)

  • Added an appendix (new Apppendix B)

---

## Round 2 · Author Response

Dear Editor,

We thank you for forwarding the Referees' reports. We are grateful to both of them for the positive assessment of our work and for the comments they made. Hereafter we respond the comments from the reports and indicate the corresponding changes whenever applicable. We hope that with the changes, the paper is now suitable for publication in SciPost Physics.

Best regards,
Tong Liu, Xu Xia, Stefano Longhi, and Laurent Sanchez-Palencia

Referee's comment/question 1 : "In the finite-size scaling analysis for the exponent beta (Fig.2a, Fig.3c, Fig.5) the authors observed a linear dependence of beta when plotted as a function of 1/m. Since the system size L=Fm, where Fm is the m-th Fibonacci number, one finds m \propto log(L) for large enough m. The finite-size correction to the exponent beta is therefore of order 1/log(L). Is this behavior general and expected for second order phase transitions or is it specific to the model considered ? "

Answer : The linear finite-scaling with the Fibonacci index is quite standard in quasiperiodic models. It holds for other models as well, including the celebrated Aubry-Andre-Harper (quasi Mathieu) Hamiltonian. However, we are not aware of studies establishing a direct relation between the 1/log(L) scaling of the corrections and the phase transition class.

Referee's comment/question 2 : "A comment to Fig.4. To my understanding, the results shown do not depend on the value of the phase because the irrational parameter alpha used in the calculation is approximated by the ratio of consecutive Fibonacci numbers. This, together with the periodic boundary conditions, makes the choice of the phase irrelevant and therefore disorder averaging can be avoided."

Answer : The argument suggested by the Referee is in principle correct and works in case of pure point or critical spectrum (as in our case). However, as a matter of fact in our simulations the energy spectra shown in Fig.4, for different and some illustrative values of the phase \theta, are obtained assuming a lattice size L=1000 and using open boundary conditions. The lattice size is thus not equal to nor proportional to any Fibonacci number and we directly use the irrational Diophantine number \alpha=(\sqrt{5}-1)/2. Yet, we could not observe any marked dependence of the spectra on \theta. This may be attributed to the irrational Diophantine nature of \alpha: this is why in the numerical computation of the eigenvalues of the matrix Hamiltonian for large L and rational approximant \alpha=p_n/q_n with q_n>L one cannot notice any marked sensitivity of the energy spectrum on \theta. Conversely, if one assume a system size L much larger than q_n, one could observe in the numerical simulations a dependence on \theta since the system becomes closer to periodic rather than aperiodic.

Referee's comment/question 3 : "From the Appendix A it seems that, if the disorder potential v(x) is unbounded and quasi-periodic, the Lyapunov exponent is independent of the specific form of v(x), as the final result coincides with the clean limit case. Is this statement correct?"

Answer : In fact, for a quasi-periodic and unbounded potential, the Lyapunov exponent depends rather generally on the particular form of v(x) and on the energy. For example, for the Maryland model, v(x)=V tan (\pi \alpha n + \theta), which is quasi-periodic and unbounded potential, the Lypaunov exponent L(E) can be computed analytically (see e.g. Ref.[50] in our manuscript) and the result is different than for our model. The main point is that the Lyapunov exponent in the "clean limit model" is the value of L(E) when the complex phase \epsilon added to the argument of the potential goes to infinity, while the Lypaunov exponent L(E) of the physical model, which we are looking for, is the value of L(E) when for $\epsilon=0$. Avila's global theory allows us to connect the two values of L(E), from \epsilon=\infty to \epsilon=0. However in order to use such a theory some strict conditions on the cocycle should be met.

---

## Round 2 · List of Changes

Requested change 1 : "In Fig5: which value of a is used and what is the corresponding value of Em".

Answer : The have added the values of the tuning parameter a and the corresponding mobility edges in the caption of the figure.

  1. The authors mentioned a "multifractal theorem". What are they referring to exactly ?

Answer : We thank the referee for raising this point. Indeed, this is an incorrect wording. There is no theorem in the mathematical sense. We have changed it to "multifractal analysis", which refers to the analysis of the scalings of the parameters \beta.

---

## Editorial Decision

published